# Cocaprins, β-Trefoil Fold Inhibitors of Cysteine and Aspartic Proteases from *Coprinopsis cinerea*

**DOI:** 10.3390/ijms23094916

**Published:** 2022-04-28

**Authors:** Miha Renko, Tanja Zupan, David F. Plaza, Stefanie S. Schmieder, Milica Perišić Nanut, Janko Kos, Dušan Turk, Markus Künzler, Jerica Sabotič

**Affiliations:** 1Department of Biochemistry and Molecular and Structural Biology, Jožef Stefan Institute, 1000 Ljubljana, Slovenia; mr914@cam.ac.uk (M.R.); dusan.turk@ijs.si (D.T.); 2Department of Biotechnology, Jožef Stefan Institute, 1000 Ljubljana, Slovenia; tanja.zupan@ijs.si (T.Z.); milica.perisic@ijs.si (M.P.N.); janko.kos@ffa.uni-lj.si (J.K.); 3Department of Biology, Institute of Microbiology, ETH Zürich, 8093 Zürich, Switzerland; david.plaza@ki.se (D.F.P.); stefanie.schmieder@childrens.harvard.edu (S.S.S.); markus.kuenzler@micro.biol.ethz.ch (M.K.); 4Faculty of Pharmacy, University of Ljubljana, 1000 Ljubljana, Slovenia

**Keywords:** protease inhibitor, cysteine protease, aspartic protease, β-trefoil fold

## Abstract

We introduce a new family of fungal protease inhibitors with β-trefoil fold from the mushroom *Coprinopsis cinerea*, named cocaprins, which inhibit both cysteine and aspartic proteases. Two cocaprin-encoding genes are differentially expressed in fungal tissues. One is highly transcribed in vegetative mycelium and the other in the stipes of mature fruiting bodies. Cocaprins are small proteins (15 kDa) with acidic isoelectric points that form dimers. The three-dimensional structure of cocaprin 1 showed similarity to fungal β-trefoil lectins. Cocaprins inhibit plant C1 family cysteine proteases with *K*_i_ in the micromolar range, but do not inhibit the C13 family protease legumain, which distinguishes them from mycocypins. Cocaprins also inhibit the aspartic protease pepsin with *K*_i_ in the low micromolar range. Mutagenesis revealed that the β2-β3 loop is involved in the inhibition of cysteine proteases and that the inhibitory reactive sites for aspartic and cysteine proteases are located at different positions on the protein. Their biological function is thought to be the regulation of endogenous proteolytic activities or in defense against fungal antagonists. Cocaprins are the first characterized aspartic protease inhibitors with β-trefoil fold from fungi, and demonstrate the incredible plasticity of loop functionalization in fungal proteins with β-trefoil fold.

## 1. Introduction

Protease inhibitors are important regulators of proteolytic activity, which plays an important role in many physiological and pathological processes. Protease inhibitors from fungi show great versatility, including unique types of inhibitory mechanisms. Protease inhibitors with the β-trefoil fold are a well characterized group of fungal protease inhibitors [1,2]. The β-trefoil fold consists of 12 β-strands folded into structurally similar units related by a pseudo-3-fold symmetry. β-strands are connected by 11 loops of varying length and composition that constitute approximately half of the molecule. Fungal protease inhibitors with β-trefoil fold inhibit different classes of proteases [3,4]. Mycospins that inhibit S1 family serine proteases are substrate-like canonical inhibitors that utilize different inhibitory reactive loops [5,6], whereas mycocypins that inhibit C1 family cysteine proteases occlude the protease’s active site by a distinct inhibitory mechanism [7]. Mycocypins, the cysteine protease inhibitors found only in higher fungi, have been described from *Clitocybe nebularis* [8], *Macrolepiota procera* [9], and *Laccaria bicolor* [10]. Inhibitors of aspartic proteases with the β-trefoil fold have not yet been described from fungi; however, a couple representatives have been identified from plants, namely potato cathepsin D inhibitor (PDI), which is a dual serine- and aspartic-protease inhibitor [11] and *Solanum lycopersicum* aspartic protease inhibitor (SLAPI) [12,13].

*Coprinopsis cinerea* is a model species of macrofungi for fruiting body development [14] and defense against antagonists [15]. RNA sequencing of *C. cinerea* fruiting bodies and vegetative mycelium challenged with bacteria and fungivorous nematodes revealed lists of differentially regulated genes putatively involved in these processes [16,17,18,19]. Among the encoded proteins, the trypsin-specific β-trefoil fold protease inhibitor cospin from fruiting bodies of *C. cinerea* was structurally and functionally characterized [6] as was the lectin CCL2 with the same fold [20,21]. Both proteins showed exceptional features, namely, that the inhibitory reactive site and the glycan binding site are located at a different position on the molecule compared to similar proteins from other higher fungi [5,20].

Here, we describe the genetic, biochemical, and functional characterization, including the three-dimensional structure, of two novel protease inhibitors from *C. cinerea* named cocaprin, *Coprinopsis cinerea* cysteine and aspartic protease inhibitor (CCP).

## 2. Results

### 2.1. Identification, Expression and Homologs of Cocaprins

Differential gene expression analysis of *C. cinerea* strain AmutBmut identified genes Okayama 7 CC1G_05298 and Okayama 7 CC1G_05299 as highly expressed in stage 1 primordia and vegetative mycelium, respectively [18] (Figure 1A). The latter gene was also strongly induced upon challenge of AmutBmut vegetative mycelium with the fungivorous nematode *Aphelenchus avenae* [19]. Sequence-based structural analysis with the SMART web resource [22] predicted that the two encoded proteins, CCP1 (protein ID 485770) and CCP2 (protein ID 441209), respectively, and their paralog CCP3 (protein ID 441207) encoded by gene Okayama 7 CC1G_05297 [23] contain an 83 ± 5 residue Ricin-type β-trefoil lectin-like domain. All three proteins lack a signal peptide for classical secretion and are, thus, predicted to be cytoplasmic. Remarkably, all three genes are arranged as a tandem within 4 kb (981721 bp-985676 bp) on scaffold 19 of the AmutBmut genome (Figure 1A). Okayama 7 CC1G_05297 is highly expressed both in vegetative mycelium and stage 1 primordia [17] (Figure 1B). Thus, the three paralogous genes have the same genomic location but differ significantly in their regulation pattern.

### 2.2. Biochemical Characterization

Cocaprins were expressed in bacteria using the pET24 vector and *E. coli* BL21(DE3). After solubilization of inclusion bodies in 8 M urea, cocaprins were purified by size-exclusion and metal-affinity chromatography. Recombinant CCP1 and CCP2 resolved on SDS-PAGE under reducing conditions as a single 21 kDa and 20 kDa band, respectively (Figure 2A), despite their expected molecular weights (including the His-tag and the first Met) of 16,176 Da and 16,501 Da. The molecular weight of untagged cocaprins was 15,079 Da for CCP1 and 15,404 Da for CCP2. Under native conditions, cocaprins ran at approximately 40 kDa (Figure 2B), indicating dimer formation under these conditions. The experimentally determined isoelectric point of the His-tagged cocaprins (Figure 2C) (at pH 4.2 for CCP1 and at pH 5.3 for CCP2) also differed from the theoretical one (at pH 4.8 for both). The inaccurate determination of molecular weight using SDS-PAGE has also been observed for other fungal β-trefoil proteins, such as cospin [5], macrocypins [9], *Clitocybe nebularis* lectin CNL [24] and *Macrolepiota procera* lectin MpL [25].

### 2.3. Crystal Structure of Cocaprin 1

CCP1 crystallized in P2_1_ space group with two molecules in the asymmetric unit (Table 1). CCP1 has a β-trefoil fold, consisting of only β-strands. The fold resembles a tree with a six-stranded β-barrel as a stem and additional three pairs of β-strands and connecting loops as the tree crown (Figure 3).

CCP1 is structurally very similar to MpL, the *Macrolepiota procera* ricin-B-like lectin (RMSD of 0.75 Å for 134 aligned residues), CNL, the *Clitocybe nebularis* ricin-B-like lectin (RMSD of 1.23 Å for 129 aligned residues), and designed symmetric trefoil proteins (PDB IDs 3PG0 and 4F43) with RMSD in the range of 1.3–1.5 Å. Similarity to other β-trefoil protease inhibitors from higher fungi is lower, for example macrocypin (PDB ID 3H6Q, RMSD 1.9 Å), clitocypin (PDB ID 3H6R, RMSD 2.0 Å), and cospin, a proteinase inhibitor from *Coprinopsis cinerea* (PDB 3N0K, RMSD 2.1 Å) (Figure 4).

We also attempted to crystallize CCP2, but expression levels were much lower and the protein tended to aggregate. Despite extensive crystallization attempts, we were unable to obtain useful crystals, so we calculated the AlphaFold2 model [26]. As expected, the AlphaFold2 model of CCP2 shows high similarity to the crystal structure of CCP1 (RMSD of 0.6 Å for 133 aligned CA atoms) with only minor differences, most of which are in loop regions despite a relatively large sequence difference (62.8% sequence identity). Overall, CCP1 and CCP2 differ in 50 residues. The differences are evenly distributed throughout the sequence (Appendix A). The largest differences are found in the loop regions, where 29 of 74 loop residues are different. Smaller differences are observed in the β-strands, where 21 of 65 residues are different. Overall, differences in buried core residues are generally conservative, mostly between small hydrophobic residues (A, I, V, L), whereas differences in surface-exposed residues are generally much larger (charge differences, replacement of hydrophilic by charged residues, etc.).

### 2.4. Cocaprins Are Cysteine Protease Inhibitors

Cocaprins inhibit plant cysteine proteases belonging to the C1 family, papain and ficain, with *K*_i_ in the low micromolar range (Table 2). They do not inhibit the cysteine protease legumain from common bean, which belongs to family C13. Cocaprins did not inhibit human cysteine proteases, cathepsins L and H. Furthermore, they showed no inhibition of serine proteases belonging to families S1 or S8.

### 2.5. Cocaprins Are Aspartic Protease Inhibitors

Cocaprins inhibit the aspartic protease pepsin with *K*_i_ in the low micromolar range (Table 2) and the aspartic protease rennin with IC50 at 44.5 μM (CCP1) and 20.5 μM (CCP2). Because these proteases derive from animals and their inhibition may not be biologically relevant, we tested the inhibition of a fungal aspartic protease, rhizopuspepsin, and cocaprins showed no inhibition. Suggestive for a function in defense, cocaprins showed specific inhibition for APR1, one of the digestive aspartic proteases from the parasitic nematode *Haemonchus contortus*, but did not inhibit a similar digestive protease PEP1 from the same organism.

### 2.6. Aspartic and Cysteine Proteases Are Not Inhibited through the Same Inhibitory Reactive Site

Based on the crystal structure, we designed mutations in the surface exposed loops, which were of sufficient length to be able to inhibit aspartic and cysteine proteases. Namely, we produced the G13E, N22R, FH32EE, and D47R mutants of CCP1 which were expressed in the same bacterial expression system (Appendix A) as inclusion bodies in very low yield (approximately 2 to 5 mg/L), except for CCP1 FH32EE, which was expressed as a soluble protein in a higher yield (32 mg/L). Their correct folding and functionality were confirmed by measuring the CD spectra (Appendix A) and inhibition of the target peptidases. Equilibrium constants were determined for the inhibition of papain and pepsin (Table 3), which indicated that the β2-β3 loop containing the N22R mutation was involved in papain inhibition because this mutant had an approximately ten-fold higher *K*_i_ value. Since the same mutant had a *K*_i_ value for pepsin inhibition comparable to that of the wild type CCP1, this suggests that the inhibitory reactive sites for aspartic and cysteine proteases are located at different positions on the protein.

### 2.7. Cocaprins Inhibit the Activity of Peptidases from C. cinerea Fruiting Bodies

To investigate whether cocaprins play an endogenous regulatory role, we analyzed the inhibition of proteolytic activities in *C. cinerea* fruiting bodies. Fruiting bodies are rich in proteolytic activity [28], and different types of proteolytic activity were also detected in gel filtration fractions of *C. cinerea* fruiting body extract (Appendix A). Those cleaving the substrate Boc-Gly-Arg-Arg-MCA, indicating C1 family cysteine peptidase activity (papain-like) or S1 family serine peptidase activity (trypsin-like), were strongly inhibited by CCP2 and weakly inhibited by CCP1. In contrast, cleavage of the substrate Suc-Ala-Ala-Pro-Phe-MCA, indicative of S1 (chymotrypsin-like) or S8 (subtilisin-like) serine peptidases, was unaffected by either. The degradation of azocasein, indicative of general proteolytic activity in the same fractions, was only very weakly inhibited by both cocaprins, suggesting that cocaprins target specific proteases. In addition, strong inhibition of endogenous aspartic peptidases that cleave the substrate MocAc-Ala-Pro-Ala-Lys-Phe-Phe-Arg-Leu-Lys-DnpNH_2_ at pH 4 was observed for both CCP1 and CCP2 (Appendix A).

### 2.8. Are Cocaprins Lectins?

Based on the sequence and structural similarity of cocaprins to fungal lectins MpL and CNL (Figure 3B and Figure 5), we used glycan microarray analysis to test the possibility of carbohydrate binding by cocaprins (Appendix A). For CCP1, very weak binding was observed on a mammalian glycan array to structures including LacNAc or polyLacNAc and for CCP2 the binding was even weaker. This indicates a potential for glycan-binding activity in cocaprins.

## 3. Discussion

Differential expression of genes encoding cytoplasmic lectins and protease inhibitors in vegetative mycelium and fruiting bodies is an economic strategy fungi use to defend different developmental stages from stage-specific predators and nutrient competitors [29]. The tandemly-arranged genes encoding the paralogous CCP1, CCP2, and CCP3 proteins differ in terms of their developmental regulation in that *ccp3* is produced in both the vegetative mycelium and stage 1 primordia, whereas *ccp1* is expressed only in vegetative mycelium and *ccp2* in fruiting bodies and upon induction with fungivorous nematodes [19]. We hypothesize that their differential regulation during development may be due to differences in the specificity of the three inhibitors toward endogenous or exogenous proteases, as suggested by the data for CCP1 and CCP2.

The mode of inhibition of cysteine proteases by β-trefoil inhibitors has been structurally well characterised [4,7]. In the crystal structure of clitocypin, a cysteine protease inhibitor from *C. nebularis*, in complex with the papain-like cysteine protease cathepsin V, two clitocypin loops occlude the catalytic cysteine residue and prevent substrate binding to the active site. Based on this knowledge, it is easy to expect that other β-trefoil cysteine protease inhibitors follow the same mechanism. Because of the low sequence similarity between these inhibitors, one cannot rely solely on sequence conservation but must examine the structural motifs in the different loops and the plausibility of the interaction with the target proteases. Therefore, we engineered several mutants in the surface-exposed loops that are long enough to inhibit cysteine proteases by occluding the active site. Only the N22 mutation in CCP1 significantly reduced its ability to inhibit cysteine proteases, suggesting that the β2-β3 loop is critical for inhibition, which is different from that of clitocypin. In clitocypin, the β1–β2 and β3–β4 loops were shown to occlude the active site, with two Gly-Gly residues playing an important role. In contrast, the CCP1 mutation in a corresponding Gly had no effect on papain inhibition. Another difference between cocaprins and mycocypins is that, unlike mycocypins, cocaprins do not inhibit asparaginyl endopeptidase (AEP, legumain), a C13 family cysteine protease. These differences suggest that cocaprins are not mycocypin-like fungal cysteine protease inhibitors, but a distinct family of dual-headed protease inhibitors that target cysteine and aspartic proteases.

To the best of our knowledge, the structure of cocaprin 1 is the second crystal structure of aspartic protease inhibitors with β-trefoil fold after the structure of Potato Cathepsin D Inhibitor (PDI). Based on the loop length and docking experiments, the authors speculated that the 17-residue-long proline-rich loop between residues 142 and 159 connecting β-strands β9 and β10 in PDI might be involved in the inhibition of cathepsin D; however, they did not confirm their hypothesis. This loop is also absent in their structure, presumably due to its flexibility and disorder [11]. The corresponding loop in CCP1 is much shorter and consists of only five residues (Val99-Ala103). It makes a rather tight turn between β strands β9 and β10 and does not protrude from the globular core of the protein, so it is unlikely to be involved in aspartic protease inhibition. Another characterized aspartic protease inhibitor with β-trefoil fold is the β-trefoil aspartic protease inhibitor from tomato (SLAPI). Although crystal structures are not available, a model has been published, and based on this model, the corresponding β9-β10 loop is even longer than in PDI, 20 residues long [12,13].

β-Trefoil proteins are extremely versatile. It has long been known that they can serve many biological functions and act as inhibitors, lectins, growth factors, etc. [30]. Recently, our groups have characterized several new β-trefoil proteins from higher fungi that exhibit either protease inhibitory or glycan binding activities, which has expanded our understanding of β-trefoil fold plasticity in protease inhibitors and lectins [4,5,21]. Despite very low sequence homology, these proteins share a common feature, namely a very stable core decorated with versatile loops. These loops provide different surface topologies for protein–protein and protein–glycan interactions, and distinct differences in amino acid sequences provide different strengths and specificities of interactions for different targets. The different positions of the reactive loops in the different β-trefoil proteins appear to be random, but this suggests that the position of the loops is not important for activity. What is likely important is the sequence and length of the loops, which provide the required topology. Similarly, chemical specificity, which is determined by multivalent interactions of amino acids across an intrinsically disordered protein region, engages in essential molecular functions [31].

We show indications that cocaprins may have lectin activity in addition to protease inhibition. Glycan microarray analysis suggested a possible function of cocaprins in glycan binding, but the suggested glycan binding specificity could not be conclusively confirmed by additional methods, including affinity chromatography and lectin-blot. Further experiments are needed to confirm or reject this possibility.

The biological function of cocaprins is unknown. Based on previous research on β-trefoil protease inhibitors and lectins, we investigated the possible endogenous role in the regulation of proteolytic activities, focusing on the activities in the fruiting bodies and the possible role in the defense against antagonists. Regarding the former function, we detected partial inhibition of endogenous proteolytic activities, probably belonging to the cysteine and aspartic proteases (Appendix A). Interestingly, CCP2, which is mainly expressed in fruiting bodies, inhibited cysteine protease-like activities more strongly, whereas CCP1, which is mainly expressed in mycelium, showed weaker inhibitory activity. However, both cocaprins inhibit aspartic- protease-like activities present in fruiting body extract to a similar extent. Regarding a potential role in defense, it is noteworthy that CCP2 expression was induced upon challenge with a fungivorous nematode [19]. However, no toxicity of the protein was detected against nematodes (*Caenorhabditis elegans*, *Caenorhabditis tropicalis*, *Pristionchus pacificus*) or dipteran insect larvae (mosquito *Aedes aegypti*) (Appendix A), all of which have been shown to be targeted by other β-trefoil protease inhibitors and lectins [10,32]. We also tested the cytotoxicity of cocaprins against several mammalian cell lines, including CaCo2, HeLa, Jurkat, human microglia (Hμglia), and U937, and cocaprins showed no cytotoxicity except for limited cytotoxicity to the CaCo2 cell line (Appendix A). This suggests a possible endogenous developmental role for cocaprins, but a defense function against other potential fungal antagonists remains possible.

## 4. Materials and Methods

### 4.1. Enzymes, Substrates and Inhibitors

Papain (2 × crystallized) (EC 3.4.22.2), ficain (EC 3.4.22.3), bovine rennin (3.4.23.4), porcine pepsin (3.4.23.1), rhizopuspepsin, bovine chymotrypsin (EC 3.4.21.1), bovine trypsin (EC 3.4.21.4), and porcine kallikrein (EC 3.4.21.35) were obtained from Sigma-Aldrich (St. Loius, MO, USA), bovine thrombin (EC 3.4.21.5) from Calbiochem (La Jolla, CA, USA), *Bacillus subtilis* subtilisin (EC 3.4.21.62) from Boehringer Mannheim (Mannheim, Germany) and porcine elastase (EC 3.4.21.36) from Serva (Heidelberg, Germany). Legumain (EC 3.4.22.34) was isolated from germinated bean seeds as previously described [33] and recombinant human cathepsin L (EC 3.4.22.43) and cathepsin H (EC 3.4.22.16) were prepared as described [9]. *Haemonchus contortus* proteases APR1 and PEP1 were prepared recombinantly in insect cells [34]. Substrates benzyloxycarbonyl(Z)-Phe-Arg-MCA (7(4-methyl)-coumarylamide), Arg-MCA, Z-Ala-Ala-Asn-MCA, Suc-Ala-Ala-Pro-Phe-MCA, Boc-Gly-Arg-Arg-MCA, H-Pro-Phe-Arg-MCA, t-butoxycarbonyl-Val-Pro-Arg-MCA, and Suc-Ala-Ala-Ala-MCA were from Bachem (Bubendorf, Switzerland), and MOCAc-Ala-Pro-Ala-Lys-Phe-Phe-Arg-Leu-Lys-(Dnp)-NH2 was from Peptide Institute Inc. (Osaka, Japan). Synthetic cysteine protease inhibitor E-64 and the aspartic protease inhibitor pepstatin A were from Peptide Institute Inc.

### 4.2. Cloning, Heterologous Expression and Purification of Recombinant Cocaprins

Total RNA extraction and cDNA synthesis of CC1G_05299 and CC1G_05298 from *C. cinerea* AmutBmut stage 1 primordia and vegetative mycelium were performed as described previously [18]. In brief, cDNA was synthesized using cDNA transcriptor universal (Roche Life Science/Merck Millipore, Burlington MA, USA) from 2 µg of total RNA following the manufacturer’s instructions. The His8-tagged version of CC1G_05299 was amplified from cDNA derived from *C. cinerea* AmutBmut vegetative mycelium, using primers CC1G_05299 Fw8HisNdeI and CC1G_05299 RvNotI (Appendix A). The PCR product was cloned into pGEM-T-easy vector (Promega, Madison WI, USA), which was then used to transform chemocompetent *E. coli* DH5α. Using the NdeI and NotI restriction sites, a mutation free insertion was cloned into the pET24b expression vector (Novagen/Merck Millipore, Burlington, MA, USA).

The His8-tagged version of CC1G_05298 was amplified from cDNA derived from *C. cinerea* AmutBmut stage 1 primordia, using the primer pair CC1G_05298 FwNdeI and CC1G_05298 Rv8HisBamHI (Appendix A). The PCR product was cloned into the pET24b expression vector (Novagen) using the restriction sites NdeI and BamHI. Correct insertion was verified by sequencing.

For protein expression, the plasmids were transformed into *E. coli* BL21 (DE3). To test protein expression and solubility, transformants were cultivated in LB medium containing 50 µg/mL kanamycin to an OD600 = 0.7 and expression was induced with 1 mM isopropyl-β-D-thiogalactopyranoside (IPTG) at 24 °C for 16 h. Solubility of the proteins was assayed as previously described [35].

In order to produce recombinant cocaprins, bacteria were collected by centrifugation (15 min, 6000 *g*, 4 °C) and sonicated in lysis buffer (50 mM Tris-HCl, 2 mM EDTA, 0.1% Triton X—100, pH 7.5). After washing the pellet with lysis buffer, the inclusion bodies were solubilized in lysis buffer containing 8 M urea. Following gel filtration on Sephacryl S200 in 20 mM Tris-HCl, 0.3 M NaCl, pH 7.5, cocaprins were purified using metal affinity chromatography with TALON^®^ Metal Affinity Resin following the protocols recommended by the manufacturer (Clontech/Takara Bio Inc., Kusatsu, Japan).

### 4.3. Mutagenesis

Mutants of CCP1 (CC1G_05299) D47R, N22R, FH32EE and G13E were produced by PCR site-directed mutagenesis (primers used are listed in Appendix A) using the appropriate pET vectors as templates followed by digestion with DpnI (Fermentas, St. Leon-Rot, Germany) and recovery of the vectors containing mutated inserts [36]. Their expression and purification were the same as for the wild type cocaprins.

### 4.4. SDS-PAGE, Native-PAGE, and Isoelectric Focusing

The proteins were routinely analyzed on 15% polyacrylamide gels under denaturing and reducing conditions, and visualized using Coomassie brilliant blue staining or silver staining. Low molecular weight markers of 14.4 kDa to 97 kDa (GE Healthcare Life Sciences, Buckinghamshire, England) were used for molecular mass estimations. The proteins were analyzed under non-denaturing conditions using blue native PAGE with a Novex NativePAGE Bis-Tris gel system with 4% to 16% gradient protein gels (ThermoFisher Scientific, Waltham, MA, USA), according to the manufacturer instructions. NativeMark unstained protein standards (ThermoFisher Scientific) was used for the molecular mass estimations. Isoelectric focusing was carried in precast Novex pH 3–10 IEF protein gels (ThermoFisher Scientific) following the manufacturer instructions. Marker proteins with pI values from 3.5 to 9.3 were used for calibration (GE Healthcare).

### 4.5. Structure Solution and Refinement

Cocaprin 1 was concentrated in 10 mM Tris-HCl, 100 mM NaCl buffer, pH 7.5 to 20 mg/mL. Crystals were grown in 0.1 M HEPES, 1.3 M Trisodium citrate, pH = 7.5. The data set was collected at the BM14 beamline (ESRF, Grenoble, France) to 1.7 Å resolution. The structure was solved with molecular replacement using Phaser [37] with poly alanine chain of MOA, a lectin from *Marasmius oreades* (PDB ID 2IHO) [38], as a search model. The whole structure was built using ArpWarp [39], combined with manual inspection and corrections in Coot [40]. The structure was finally refined with Refmac [41] and deposited to PDB with the PDB ID 7ZNX.

### 4.6. Inhibition Assays and Determination of Kinetic Constants

Inhibition kinetics were determined under pseudo-first order conditions and analyzed according to Henderson [27] as described [6]. Kinetic assays were performed for the cysteine proteases papain, ficain and cathepsin L using Z-Phe-Arg-MCA as substrate in 0.1 M MES buffer, pH 6.5 with 5 mM DTT, for cathepsin H using Arg-MCA in 0.1 M MES buffer with 1.5 mM EDTA and 2 mM DTT and for legumain using Z-Ala-Ala-Asn-MCA substrate in 0.1 M Na-acetate buffer, pH 5.5 with 5 mM DTT. Aspartic proteases pepsin, rennin, rhizopuspepsin, *Hc*APR1, and *Hc*PEP1 were assayed in 0.1 M phosphate-citrate buffer, pH 3.4 using MocAc-Ala-Pro-Ala-Lys-Phe-Phe-Arg-Leu-Lys-DnpNH2 as substrate. Trypsin was assayed using the substrate Boc-Gly-Arg-Arg-MCA in 0.1 M Tris-HCl buffer, pH 8.0, containing 20 mM CaCl_2_. Chymotrypsin and subtilisin were assayed in 0.1 M Tris-HCl buffer, pH 8.8 with Suc-Ala-Ala-Pro-Phe-MCA as substrate. Elastase was assayed in 0.1 M Tris-HCl buffer, pH 8.0, containing 20 mM CaCl_2_ with Suc-Ala-Ala-Ala-MCA. Kallikrein was assayed in 0.05 M Tris-HCl, 0.05 M NaCl, 0.01% (*v*/*v*) Tween with H-Pro-Phe-Arg-MCA as substrate and thrombin was assayed in 0.25 M phosphate buffer, pH 6.5, with the substrate t-butoxycarbonyl-Val-Pro-Arg-MCA. The released MCA was measured using an Infinite^®^M1000 microplate reader (Tecan, Männedorf, Switzerland).

### 4.7. Glycan Microarray Analysis

Tests on the mammalian printed glycan array, version 5.2) (http://www.functionalglycomics.org/static/consortium/resources/resourcecoreh8.shtml, accessed 14 January 2022) with 609 glycans were conducted by the Consortium for Functional Glycomics (Protein–Glycan Interaction Core, formerly Core H) as described previously [24,42]. Recombinant cocaprins were biotinylated using No-WeightTM NHS-PEO4-Biotin (Pierce, Rockford, IL, USA) in accordance with the manufacturer’s instructions. Binding of cocaprins (at 200 μg/mL) to the array was detected by streptavidin Alexa Fluor 488 conjugate. The highest and the lowest results from each set of replicates were removed to eliminate false hits, and average binding (rank) was calculated.

## 5. Conclusions

The characterization of cocaprins as a new family of unique protease inhibitors from higher fungi highlights the incredible reservoir of β-trefoil fold diversity in fungi. Cocaprins are the first example of β-trefoil aspartic protease inhibitors from higher fungi and provide valuable information about this elusive group of proteins. Indeed, only a handful of aspartic protease inhibitors have been characterized to date, whereas there are many aspartic proteases known to be involved in pathogenic processes [43], warranting further efforts in the search and characterization of new aspartic protease inhibitors.

## Figures and Tables

**Figure 1 ijms-23-04916-f001:**
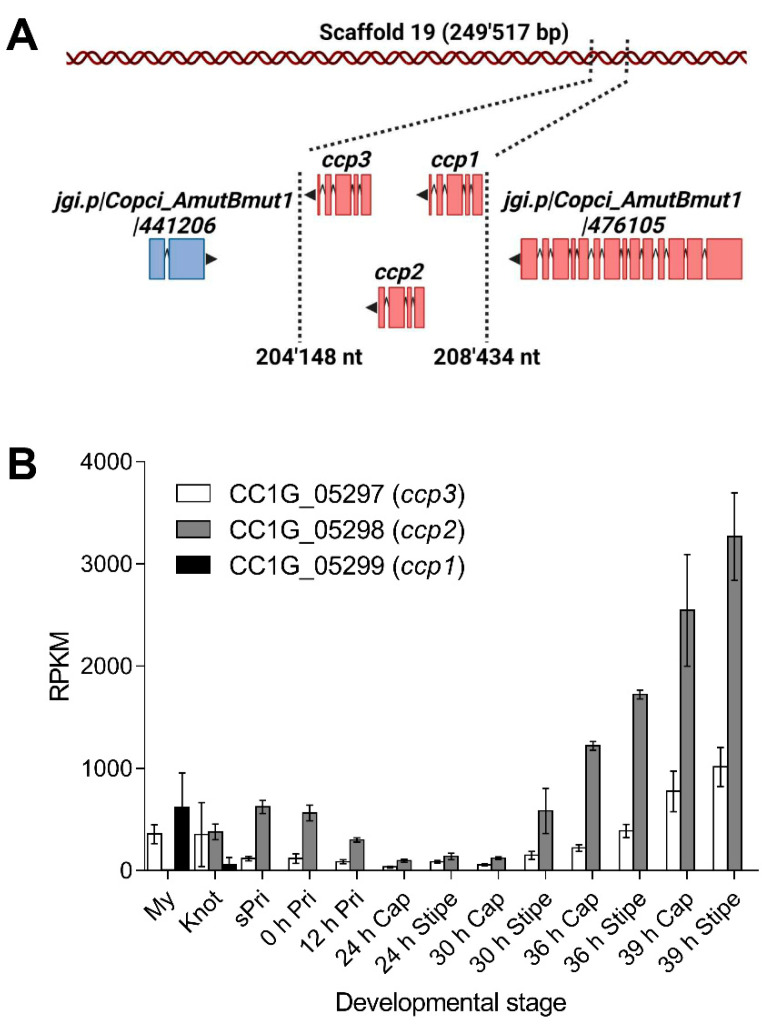
Cocaprin-encoding genes form a tandem array in the genome of *C. cinerea* strain AmutBmut and show developmental regulation. (**A**) Representation of the cocaprin locus on scaffold 19 with arrowhead and colors indicating the direction of transcription. Boxes show the location of exons; (**B**) Mean RPKM (Reads Per Kilobase of transcript per Million mapped reads, *n* = 2) corresponding to 13 developmental stages in *C. cinerea* strain AmutBmut for *ccp1*, *ccp2* and *ccp3* [17]. Standard deviations are shown as error bars. My: Vegetative mycelium, Knot: Hyphal knots with vegetative mycelium, sPri: Small fruiting body primordia, 0 h Pri: Fruiting body primordia at 0 h, 12 h Pri: Fruiting body primordia at 12 h after the trigger light, 24 h Cap: Cap at 24 h after the trigger light, 24 h Stipe: Stipe at 24 h after the trigger light, 30 h Cap: Cap at 30 h after the trigger light, 30 h Stipe: Stipe at 30 h after the trigger light, 36 h Cap: Cap at 36 r after the trigger light, 36 h Stipe: Stipe at 36 h after the trigger light, 39 h Cap: Cap at 39 h after the trigger light and 39 h Stipe: Stipe at 39 h after the trigger light.

**Figure 2 ijms-23-04916-f002:**
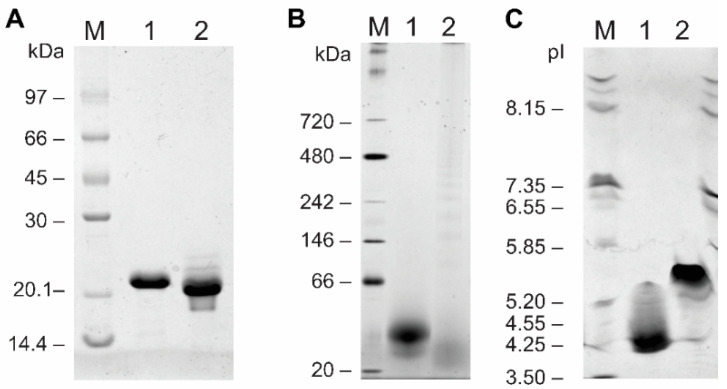
Polyacrylamide electrophoretic analyses of CCP1 and CCP2. (**A**) SDS-PAGE analysis in a 15% polyacrylamide gel under denaturing conditions showing Coomassie blue staining; (**B**) blue native PAGE analysis; (**C**) isoelectric focusing analysis. Lane M: appropriate standard protein mix; lane 1: CCP1; lane 2: CCP2.

**Figure 3 ijms-23-04916-f003:**
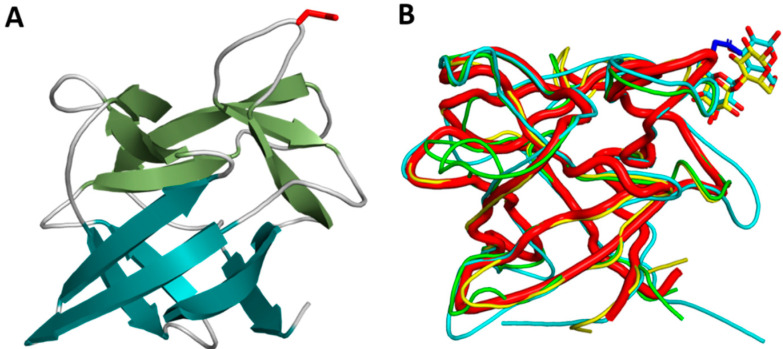
Three-dimensional structure of cocaprin 1. (**A**) Crystal structure of CCP1 (7ZNX). β-strands in the trunk shown in teal and β-strands in the crown shown in green. The N22 residue involved in the inhibition of cysteine protease is shown as red sticks; (**B**) structural superposition of CCP1 (red thick ribbons) with MpL (yellow), CNL (cyan) and the designed trefoil protein (green). The lactose molecules in CNL and MPL (shown as sticks) bind in the same region as the cysteine proteases in CCP1.

**Figure 4 ijms-23-04916-f004:**
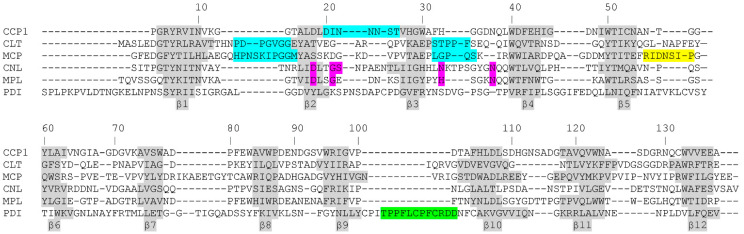
Structural alignment of cocaprins with mycocypins and selected lectins. The β-strands are shown on a gray background, and the loops involved in inhibition of cysteine proteases are highlighted in cyan, those involved in inhibition of trypsin or legumain in yellow, putative loops involved in inhibition of cathepsin D in green and glycan binding residues are highlighted in pink. CCP1, cocaprin (7ZNX), Clt, clitocypin (3H6R), Mcp, macrocypin (3H6Q), CNL, *Clitocybe nebularis* lectin (3NBD), Mpl, *Macrolepiota procera* lectin (4ION), PDI, Potato Cathepsin D Inhibitor (5DZU).

**Figure 5 ijms-23-04916-f005:**
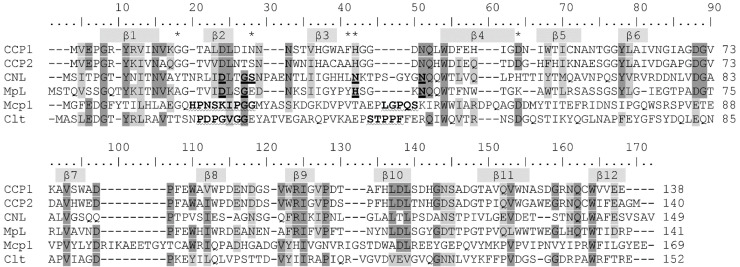
Alignment of amino acid sequences of cocaprins with selected lectins and mycocypins. Identical amino acid residues are shaded dark gray and similar ones are shaded light gray. The β-strands of cocaprin secondary structure are indicated by gray bars above the amino acid sequence. Glycan-binding residues of CNL and MpL are indicated in bold and underlined, and loops involved in inhibition of cysteine proteases by Clt and Mcp1 are indicated in bold and underlined with a wavy line. Asterisks indicate residues that were mutated in this study. The sequences were aligned with the BLOSUM62 matrix.

**Table 1 ijms-23-04916-t001:** Crystallization data collection and refinement statistics. Statistics for the highest-resolution shell are shown in parentheses.

	CCP1
Resolution range	34.6–1.60 (1.66–1.60)
Space group	P 1 21 1
Unit cell	34.7 94.4 52.2 90 103.1 90
Total reflections	310,224 (19,426)
Unique reflections	41,795 (3485)
Multiplicity	7.4 (5.6)
Completeness (%)	97.2 (81.7)
Mean I/sigma(I)	26.4 (2.8)
Wilson B-factor	12.22
R-merge	0.062 (0.58)
R-meas	0.066 (0.64)
R-pim	0.024 (0.2642)
CC1/2	0.999 (0.831)
CC*	1 (0.953)
Reflections used in refinement	39,704 (3484)
Reflections used for R-free	2090 (174)
R-work	0.163 (0.220)
R-free	0.201 (0.222)
CC(work)	0.965 (0.891)
CC(free)	0.950 (0.902)
Number of non-hydrogen atoms	2494
macromolecules	2111
ligands	0
solvent	383
Protein residues	272
RMS(bonds)	0.014
RMS(angles)	1.76
Ramachandran favored (%)	98.88
Ramachandran allowed (%)	1.12
Ramachandran outliers (%)	0.00
Rotamer outliers (%)	1.40
Clashscore	1.74
Average B-factor	15.59
macromolecules	13.45
solvent	27.40

**Table 2 ijms-23-04916-t002:** Inhibitory pattern of cocaprins. Equilibrium constants (*K*_i_) for the inhibition of papain, ficain and pepsin were determined according to Henderson [27]. IC50 values are marked with astersks on both sides and indicated for rennin and APR1. Experiments were performed at 30 °C. S.D. are given where appropriate; NI, no inhibition.

Protease	Protease Family	Source Organism	Inhibition (*K*_i_ [µM] or *IC50* [µM])	
			Cocaprin 1	Cocaprin 2
papain	C1	*Carica papaya*	5.63 ± 2.62	16.25 ± 2.49
ficain	C1	*Ficus glabrata*	2.09 ± 0.18	1.19 ± 0.12
cathepsin L	C1	*Homo sapiens*	NI	NI
cathepsin H	C1	*Homo sapiens*	NI	NI
legumain	C13	*Phaseolus vulgaris*	NI	NI
pepsin	A1	*Sus scrofa*	0.86 ± 0.20	0.34 ± 0.11
rennin	A1	*Bos taurus*	*44.5*	*20.5*
rhizopuspepsin	A1	*Rhizopus* sp.	NI	NI
PEP1		*H. contortus*	NI	NI
APR1		*H. contortus*	*30.7*	*25.0*
trypsin	S1	*Bos taurus*	NI	NI
chymotrypsin	S1	*Bos taurus*	NI	NI
thrombin	S1	*Bos taurus*	NI	NI
elastase	S1	*Sus scrofa*	NI	NI
kallikrein	S1	*Sus scrofa*	NI	NI
subtilisin	S8	*Bacillus subtilis*	NI	NI

**Table 3 ijms-23-04916-t003:** Inhibition constants of papain and pepsin by CCP1 mutants. Equilibrium constants for the inhibition of papain and pepsin were determined according to Henderson [27]. Experiments were performed at 30 °C. S.D. are given where appropriate; NI, no inhibition.

Protease	Inhibition (*K*_i_ [µM])				
	WT	G13E	N22R	FH32EE	D47R
Papain (C1)	5.63 ± 2.62	5.22 ± 0.40	48.04 ± 11.99	7.90 ± 4.04	1.22 ± 0.32
Pepsin (A1)	0.86 ± 0.20	0.61 ± 0.37	0.33 ± 0.16	0.45 ± 0.22	1.83 ± 0.86

## Data Availability

The structure and diffraction data were deposited to PDB. All other data are contained within the article or Appendix A.

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
