# Peer review of "Cocaprins, β-Trefoil Fold Inhibitors of Cysteine and Aspartic Proteases from Coprinopsis cinerea"

_ijms, 2022, doi:10.3390/ijms23094916_

Round 1

Reviewer 1 Report

I have evaluated the paper "Cocaprins, ß-trefoil fold inhibitors of cysteine and aspartic proteases from Coprinopsis cinera" wherein the authors describe the identification, cloning, expression, purification, and structural characterization of a new class of small fungal protease inhibitor from the abovementioned mushroom species. Overall the manuscript is sound and the biochemical experiments performed are appropriate.

I have a few things that I think the authors could improve and/or clarify:

Fig 2. & supplementary Fig. 1 - CCP1 and 2 clearly purify at ~20-22 kD, but the authors note that they expected a 16 kD protein. The mutant variants shown in supplementary Fig. 1 are clearly within the expected range. Why the discrepancy? I don't see any attempt to explain this.

Table 2 - Why are the inhibitor Ki values not reported in the table for rennin and APR1 not reported here? They are reported in the text, but it would also make sense to report them here.

Section 2.6 - Is it possible to speculate where the inhibitory reactive site may be for the cysteine protease?

Line 209 - vawy should be wavy.

Supplementary Fig. S4. The caption needs to be more descriptive and clearly state what each sub-figure is showing, much as the other captions do.

Reviewer 2 Report

The authors describe the characterisation of two proteins from the fungus Coprinopsis cinerea, which they had previously identified. They caracterize their activity as inhibitors of two protease types, the plant C1 family cysteine proteases, and the aspartic protease pepsin. Mutations have been performed to identify regions involved in this anti-protease activity.

No reason is given why the only the first protein has been studied structurally. The authors should have at least provided a model calculated by the alphafold2 server for the second protein. Legend of Figure 2 is not complete (authors should detail lane 1 and 2).

What are the main difference in sequences between both proteins? are they located in loops or beta-strands? A figure comparing sequence difference mapped onto the structure would be very informative.

Finally, as the proteins has been purified in urea, a profil of gel filtration after refolding should be included, to judge the quality of refolding.
